# Path-Tracking of a WMR Fed by Inverter-DC/DC Buck Power Electronic Converter Systems

**DOI:** 10.3390/s20226522

**Published:** 2020-11-15

**Authors:** Victor Manuel Hernández-Guzmán, Ramón Silva-Ortigoza, Salvador Tavera-Mosqueda, Mariana Marcelino-Aranda, Magdalena Marciano-Melchor

**Affiliations:** 1Facultad de Ingeniería, Universidad Autónoma de Querétaro, Querétaro 76010, Mexico; 2Laboratorio de Mecatrónica & Energía Renovable, Centro de Innovación y Desarrollo Tecnológico en Cómputo, Instituto Politécnico Nacional, Ciudad de México 07700, Mexico; rsilvao@ipn.mx (R.S.-O.); staveram1500@alumno.ipn.mx (S.T.-M.); mmarciano@ipn.mx (M.M.-M.); 3Sección de Estudios de Posgrado e Investigación, Unidad Profesional Interdisciplinaria de Ingeniería y Ciencias Sociales y Administrativas, Instituto Politécnico Nacional, Ciudad de México 08400, Mexico; mmarcelino@ipn.mx

**Keywords:** energy-based control, wheeled mobile robots, path-tracking, inverter-DC/DC Buck power converter system, lyapunov stability

## Abstract

This paper is concerned with path-tracking control of a wheeled mobile robot. This robot is equipped with two permanent magnet brushed DC-motors which are fed by two inverter-DC/DC Buck power converter systems as power amplifiers. By taking into account the dynamics of all the subsystems we present, for the first time, a formal stability proof for this control problem. Our control scheme is simple, in the sense that it is composed by four internal classical proportional-integral loops and one external classical proportional-derivative loop for path-tracking purposes. This is the third paper of a series of papers devoted to control different nonlinear systems, which proves that the proposed methodology is a rather general approach for controlling electromechanical systems when actuated by power electronic converters.

## 1. Introduction

Pulse width modulation (PWM)-based power amplifiers are commonly employed to supply power to electromechanical systems. However, the intrinsic hard commutation of PWM strategies stresses the electromagnetic components of electric motors [1]. This has motivated the use of DC/DC power electronic converters instead of PWM-based power amplifiers to control DC-motors [1,2,3,4,5,6,7,8,9,10,11,12,13,14,15,16,17,18,19,20,21,22]. An important exception is [23] which presented, for the first time, a solution for control of a magnetic levitation system that is fed by a DC/DC Buck power electronic converter. The importance of this application is that magnetic levitation systems are nonlinear electromechanical systems.

Despite these advancements, the above solutions are constrained to unidirectional control, i.e., when power amplifier is only required to provide a single polarity voltage. This has motivated a series of works [24,25,26] where, by cascade connecting an inverter and a DC/DC power electronic converter, bidirectional velocity control in DC-motors has been rendered possible. Moreover, the use of this converter topology has been extended in [27] to control velocity in a permanent magnet synchronous motor, which is well known to be a nonlinear alternating current (AC) motor.

On the other hand, wheeled mobile robots (WMR) represent another class of nonlinear and nonholonomic systems. One important control problem in this field is path-tracking. This task is accomplished when driving a mobile robot as close as possible to a previously defined reference path which is specified as a set of straight lines or arcs of circumferences [28]. Several control schemes were proposed in the literature. In [29] is presented a predictive controller based on the robot kinematic model. A backstepping controller based on the kinematic model is proposed in [30] which takes into account robot skidding and slipping. A Kalman-based active observer controller (AOB) designed on the basis of both the kinematic and dynamic models of robot is proposed in [31] in order to take into account the model uncertainties. Finally, in [32] is presented a Fuzzy-logic controller based on both the kinematic and the dynamic models of robot.

In industrial applications, the path-tracking control problem is solved using a multi-loop control strategy. The most external loop (a classical proportional-integral-derivative (PID) controller) is used to control the path-tracking error. Output of this controller represents the desired velocity that the wheels of robot must track in order to accomplish the task. Inner velocity loops, driven by classical proportional-integral (PI) controllers, are used to compute the desired torques that must be generated by motors to force the actual wheel velocities to reach their desired values [33]. Moreover, another internal loop, driven again by classical PI controllers, is employed to compute voltage to be applied to motors in order to ensure that generated torques reach their desired values.

Although the work in [28] is the only one in the literature using a standard PID controller. However, that proposal is based only on the kinematic model of robot. Furthermore, any of the additional internal PI control loops that are cited above are not studied although they are present in the commercial robot that they use in the experiments.

The main contributions of the present paper are the following.

We extend the application of the control technique employed in [23,27], to the case when path-tracking is performed by a WMR that employs two inverter-DC/DC Buck power electronic converter systems to feed both brushed DC-motors used as actuators. This is the first time that a formal stability proof is presented for this control problem by considering the dynamics of all the components together.On the other hand, the present paper demonstrates that our control design method can be applied to control several different nonlinear electromechanical systems and, hence, it represents a rather general methodology. Moreover, we stress that our approach has allowed us to extend the solution for complex nonlinear electromechanical systems that are fed by DC/DC power electronic converters. This feature is very important to be highlighted since, contrary to our previous proposals in [23,27] and the present proposal, all proposals in the literature are constrained to control simple DC motors with simple linear loads. See for instance [1,2,3,4,5,6,7,8,9,10,11,12,13,14,15,16,17,18,19,20,21,22].Our proposal also constitutes an extension of the work in [33] to the case when the electrical dynamics of motors and dynamics of the inverter-DC/DC Buck power electronic converters are considered together with kinematic and dynamic models of WMR.Control scheme that we propose in the present paper is composed by four internal PI control loops and one external PD path-tracking loop, which results in a simple and robust control law which is very similar to that commonly used in industrial practice. We stress that different PID control schemes have been proposed previously in the literature to solve the path-tracking problem [28]. However, stability of the external-PD (for tracking error) internal-PI (for motor velocities) control scheme that is standard in industrial applications has been studied only in [33] and any other work on this subject has not been presented in the automatic control literature nor the robotics literature. The contribution of the present paper with respect to [33] is that we take into account the electrical dynamics of both DC motors and dynamics of the power electronic converters that are used as power amplifiers. We stress that this is a novel control problem in wheeled mobile robots and this explains why any references are not found in the scientific literature.We present a formal stability proof ensuring the state boundedness and ultimate boundedness. We stress that the ultimate boundedness result is consistent with current practice. This is because changes in the path direction actuate as time varying disturbances avoiding asymptotic convergence to an equilibrium point. See Remark 3.

The key for results in [23,27] and in the present paper is a novel passivity-based approach possessing the following features.

It exploits, in a novel and advantageous manner, the energy exchange that naturally exists among the inverter-DC/DC Buck power electronic converter system, DC-motors and WMR. Contrary to the approach in [34], we do not require to include additional terms in the control law in order to ease the achievement of such cancellations. On the other hand, although backstepping is another control technique that it is commonly employed for electromechanical systems [35], the resulting control laws are commonly much more complex (and, hence, sensitive to numerical errors and noise amplification) than those obtained using passivity-based approaches.Contrary to standard passivity-based approaches as that in [34], we do not rely in a nested-passivity-based control. This means that we do not require to force the electric current error to converge exponentially to zero in order to consider it as a vanishing perturbation for the mechanical subsystem. This provides our approach with the following advantages:-We avoid the necessity to feedback the time derivative of either the desired electric currents nor the desired voltage.-We include PI controllers. In this respect, we stress that the approach in [34] is forced to resort to disturbance estimators instead of PI controllers.-We do not have the need to include velocity filtering.-As we explained above, in applications as the one in the present paper, the electric current error does not converge to zero.

Finally, some remarks on nomenclature. Given a n×n symmetric matrix *A*, symbols $\lambda_{\min}\left\{A\right\}$ and $\lambda_{\max}\left\{A\right\}$ stand for the smallest and the largest eigenvalues of *A*, respectively. Given an scalar *a*, symbol |a| stands for the absolute value. Given some $x\in R^{n}$ and a n×n matrix *A*, symbols $∥x∥=\sqrt{x^{T}x}$ and ∥A∥ stand for the Euclidean norm of a vector and the spectral norm of a matrix, respectively.

This paper is organized as follows. In Section 2 we introduce the dynamical model of WMRs and that of the inverter-DC/DC power converters employed to provide power to robot motors. In Section 3 we explain how passivity properties of this model can be exploited to render simple the controller that is designed. Our proposal is presented in Section 4. In Section 5 we present some simulations to give some insight on the achievable performance. Finally, some concluding remarks are given in Section 6.

## 2. The Dynamical Model to be Considered

### 2.1. Wheeled Mobile Robot Model

The dynamical model of WMR that we consider (see Figure 1) has been introduced in [36]. The following nomenclature was introduced in the above paper but, for the ease of reference, we repeat it here. It is a mobile robot with two actuated wheels. Robot width is 2b whereas *r* is radius of each wheel. *O* is the xy world coordinate system whereas $P_{0}$ is the XY coordinate system fixed to WMR. Origin of $P_{0}$ is located at the middle of axis that is common to both driving wheels. $P_{c}$ is center of mass of WMR, which is on the X–axis, at a distance *d* from $P_{0}$. Mass of the body and wheel with a motor are $m_{c}$ and $m_{w}$, respectively, $I_{c}$ is the moment of inertia of the body about the vertical axis through $P_{c}$, whereas $I_{w}$ and $I_{m}$ are inertias of the wheel with a motor about the wheel axis, and the wheel with a motor about the wheel diameter, respectively.

The generalized coordinates of WMR are $q={\left[x,y,\phi ,\theta_{r},\theta_{l}\right]}^{T}$, where (x,y) are the Cartesian coordinates of $P_{0}$, ϕ is the WMR heading angle, whereas $\theta_{r}$, $\theta_{l}$ are the angles of the right and left driving wheels. We assume that wheel slipping is not present. The dynamic model is given as [36]: (1)$$\begin{array}{ccc} \dot{q} & = & S(q)\nu (t), \end{array}$$(2)$$\begin{array}{ccc} \bar{M}\left(q\right)\dot{\nu }+\bar{C}\left(q,\dot{q}\right)\nu & = & \bar{B}\left(q\right)\tau , \end{array}$$
where: (3)$$\begin{array}{ccc} S(q) & = & \left[\begin{array}{cc} \frac{r}{2}\cos\phi & \frac{r}{2}\cos\phi \\ \frac{r}{2}\sin\phi & \frac{r}{2}\sin\phi \\ \frac{r}{2b} & -\frac{r}{2b}\\ 1 & 0\\ 0 & 1 \end{array}\right],\\ \bar{M}\left(q\right) & = & \left[\begin{array}{cc} \frac{r^{2}}{4b^{2}}\left(mb^{2}+I_{0}\right)+I_{w} & \frac{r^{2}}{4b^{2}}\left(mb^{2}-I_{0}\right)\\ \frac{r^{2}}{4b^{2}}\left(mb^{2}-I_{0}\right) & \frac{r^{2}}{4b^{2}}\left(mb^{2}+I_{0}\right)+I_{w} \end{array}\right],\\ \bar{C}\left(q,\dot{q}\right) & = & \left[\begin{array}{cc} 0 & \frac{r^{2}}{2b}m_{c}d\dot{\phi }\\ -\frac{r^{2}}{2b}m_{c}d\dot{\phi } & 0 \end{array}\right],\;\bar{B}\left(q\right)=\left[\begin{array}{cc} 1 & 0\\ 0 & 1 \end{array}\right], \end{array}$$$\tau ={\left[\tau_{r},\tau_{l}\right]}^{T}$ represents torques applied at the right and the left wheels, i.e., $\tau_{r}$ and $\tau_{l}$, respectively. Angular velocities of the right ($\nu_{1}$) and the left ($\nu_{2}$) wheels are grouped in $\nu ={\left[\nu_{1},\nu_{2}\right]}^{T}$, whereas $m=m_{c}+2m_{w}$ and $I_{0}=m_{c}d^{2}+2m_{w}b^{2}+I_{c}+2I_{m}$. Finally, if $v=\sqrt{{\dot{x}}^{2}+{\dot{y}}^{2}}$, $w=\dot{\phi }$ are defined, we obtain from (Equation 1): (4)$$\begin{array}{ccc} \frac{d}{dt}\left[\begin{array}{c} x\\ y\\ \phi \end{array}\right] & = & \left[\begin{array}{cc} \cos\phi & 0\\ \sin\phi & 0\\ 0 & 1 \end{array}\right]\left[\begin{array}{c} v\\ w \end{array}\right], \end{array}$$(5)$$\begin{array}{ccc} \left[\begin{array}{c} v\\ w \end{array}\right] & = & A\left[\begin{array}{c} \nu_{1}\\ \nu_{2} \end{array}\right],\;A=\left[\begin{array}{cc} \frac{r}{2} & \frac{r}{2}\\ \frac{r}{2b} & -\frac{r}{2b} \end{array}\right]. \end{array}$$

We stress that *A* is a nonsingular matrix.

To take into account the electrical dynamics of the permanent magnet brushed DC-motors, model (1), (2), must be rewritten as follows: (6)$$\begin{array}{ccc} \dot{q} & = & S(q)\nu (t), \end{array}$$(7)$$\begin{array}{ccc} \bar{M}\left(q\right)\dot{\nu }+\bar{C}\left(q,\dot{q}\right)\nu +f\nu & = & \bar{B}\left(q\right)nk_{m}I, \end{array}$$(8)$$\begin{array}{ccc} L_{a}\dot{I} & = & -RI-nk_{m}\nu +V. \end{array}$$

Both DC-motors are assumed to be identical with gearboxes reduction rate *n*, whereas $L_{a},R,k_{m},f$, are positive constants representing armature inductance, resistance, torque constant and viscous friction coefficient. The variables $I,V\in R^{2}$ represent electric current in both motors and voltage applied at the motors armature terminals, respectively. Finally, as explained above, the motor mechanical parameters are included in both $\bar{M}\left(q\right)$ and $\bar{C}\left(q,\dot{q}\right)$. We take advantage from the fact that torque constant and back electromotive force constant are the same in permanent magnet brushed DC-motors.

### 2.2. Mathematical Model of the Inverter-DC/DC Buck Power Electronic Converter-DC Motor System

The inverter-DC/DC Buck power electronic converter-DC motor system that was introduced in [24] is depicted in Figure 2. The following description has been introduced in the above paper. Moreover, this description is also included in [27] because the same inverter-DC/DC Buck power electronic converter system is employed but for control a different electromechanical system. We repeat here the following description for the ease of reference and because we use the same inverter-DC/DC Buck power electronic converter-DC motor system introduced in [24]. We use subindex j=1,2, to designate the inverter-DC/DC Buck power electronic converter-DC motor system for the right and left motors, respectively. We have four transistors $Q_{1j},Q_{2j},{\bar{Q}}_{1j},{\bar{Q}}_{2j}$, an inductor *L*, a capacitor *C* and a resistance $R_{c}$. The arrangement of components presented Figure 2 is repeated two times to drive two different DC-motors. Symbols $i_{cj}$, $V_{j}$, $I_{j}$, represent the electric current through the inductance *L*, voltage at the capacitor *C* terminals, and electric current through the j-th DC-motor. The symbol *E* stands for voltage of the DC power supply. The system inputs are $v_{j}$, which only take the discrete values {+1,−1,0} representing the on-positive, the on-negative and the discharging states of transistors $Q_{1j},Q_{2j},{\bar{Q}}_{1j},{\bar{Q}}_{2j}$ [24].

Following [24] we find that the dynamical model of two inverter-DC/DC Buck power converter systems is given as: (9)$$\begin{array}{ccc} L\frac{di_{cj}}{dt} & = & -V_{j}+Ev_{j}, \end{array}$$(10)$$\begin{array}{ccc} C\frac{dV_{j}}{dt} & = & i_{cj}-I_{j}-\frac{V_{j}}{R_{c}}, \end{array}$$
where j=1,2. Let $U={\left[u_{1},u_{2}\right]}^{T}$ represent the average values of ${\left[v_{1},v_{2}\right]}^{T}$. Also, with some abuse of notation, let $I_{c}={\left[i_{c1},i_{c2}\right]}^{T}$, $I={\left[I_{1},I_{2}\right]}^{T}$, and $V={\left[V_{1},V_{2}\right]}^{T}$, represent the average values of the corresponding variables. Thus, the average model of the above switched dynamical model can be written as: (11)$$\begin{array}{ccc} L{\dot{I}}_{c} & = & -V+EU, \end{array}$$(12)$$\begin{array}{ccc} C\dot{V} & = & I_{c}-I-\frac{1}{R_{c}}V. \end{array}$$

The complete dynamical model of system to be controlled in this paper is given by (6)–(8), (11) and (12).

**Problem statement**: Figure 3 depicts the control problem to be solved. Let $\phi^{*}$ represent the desired heading angle, where $L_{0}$ is a positive constant. Suppose that ϕ(0) is close to $\phi^{*}\left(0\right)$. Design an input $U\in R^{2}$ such that the tracking error $\phi -\phi^{*}$ has an ultimate bound.

**Assumption** **1.**
*If the tracking error $\phi -\phi^{*}$ has a small ultimate bound and $L_{0}$ is less than the robot length, then Cartesian position does not matter to achieve path-tracking, i.e., dynamics $\left(\dot{x},\dot{y}\right)$ can be neglected.*


The following result is important for our purposes and we repeat it here for the ease of reference.

**Theorem 1.** (Theorem 4.18 in ([37], p. 172)). *Let $D\subset R^{n}$ be a domain that contains the origin and V:[0,∞)×D→R be a continuously differentiable function such that:*
(13)$$\begin{array}{ccc} \alpha_{1}\left(∥x∥\right) & \leq & V\left(t,x\right)\leq \alpha_{2}\left(∥x∥\right), \end{array}$$
(14)$$\begin{array}{ccc} \frac{\partial V}{\partial t}+\frac{\partial V}{\partial x}f\left(t,x\right) & \leq & -W_{3}\left(x\right),\;\forall ∥x∥\geq \mu >0, \end{array}$$*∀t≥0 and ∀x∈D, where $\alpha_{1}$ and $\alpha_{2}$ are class K functions and $W_{3}\left(x\right)$ is a continuous positive definite function. Take r>0 such that $B_{r}\subset D$ and suppose that:*
(15)$$\begin{array}{c} \mu <\alpha_{2}^{-1}\left(\alpha_{1}\left(r\right)\right). \end{array}$$

*Then, there exists a class KL function β and for every initial state $x(t_{0})$, satisfying $∥x\left(t_{0}\right)∥\leq \alpha_{2}^{-1}\left(\alpha_{1}\left(r\right)\right)$, there is T≥0 (dependent on $x(t_{0})$ and μ) such that the solution of $\dot{x}=f\left(t,x\right)$ satisfies:*
(16)$$\begin{array}{ccc} ∥x(t)∥ & \leq & \beta (∥x\left(t_{0}\right)∥,t-t_{0}),\;\forall t_{0}\leq t<t_{0}+T, \end{array}$$
(17)$$\begin{array}{ccc} ∥x(t)∥ & \leq & \alpha_{1}^{-1}\left(\alpha_{2}\left(\mu \right)\right),\;\forall t\geq t_{0}+T. \end{array}$$

*Moreover, if $D=R^{n}$ and $\alpha_{1}$ belongs to class $K_{\infty }$, then (Equation 16) and (17) hold for any initial state $x(t_{0})$, with no restriction on how large μ is.*


Roughly speaking, Theorem 1 states that *x* converges into a domain whose radius is bounded by $\alpha_{1}^{-1}\left(\alpha_{2}\left(\mu \right)\right)$ which is known as the ultimate bound.

Given some n×n matrices *F* and *G*, with the former symmetric, and some $y,x\in R^{n}$ then ([38], p. 176), ([39], p. 26): (18)$$\begin{array}{c} \lambda_{\min}{\left\{F\right\}∥y∥}^{2}\leq y^{T}Fy\leq \lambda_{\max}\left\{F\right\}{∥y∥}^{2},\;\forall y\in R^{n}, \end{array}$$(19)$$\begin{array}{c} \pm y^{T}Gx\leq |y^{T}Gx|\leq ∥y∥\;∥G∥\;∥x∥,\;\forall y,x\in R^{n}. \end{array}$$

## 3. Open Loop Energy Exchange

In this section we explain how the plant passivity property is employed to simplify the controller design task. Similar results have been shown to stand also in control of magnetic levitation systems [23] and permanent magnet synchronous motors [27]. Hence, the ideas that we present in the following must not be seen just as a monotonic repetition of results in the previous papers. On the contrary, it must be seen as evidence of a powerful methodology that is useful to solve control problems in different plants.

Consider the dynamical model in (7), (8), $\dot{\phi }=w$, (5), (11) and (12). The total energy stored in the system is given as:(20)$$V_{e}\left(V,I_{c},I,\nu \right)=\frac{C}{2}{∥V∥}^{2}+\frac{L}{2}∥I_{c}{∥}^{2}+\frac{1}{2}L_{a}{∥I∥}^{2}+\frac{1}{2}\nu^{T}\bar{M}\left(q\right)\nu .$$

The term $\frac{C}{2}{∥V∥}^{2}$ stands for electric energy stored in capacitors of the Buck power converters, whereas $\frac{L}{2}{∥I_{c}∥}^{2}$ represents the magnetic energy stored in inductances of the Buck power converters, and $\frac{1}{2}L_{a}{∥I∥}^{2}$ stands for the magnetic energy stored in the electrical subsystem of the DC-motors. Finally, $\frac{1}{2}\nu^{T}\bar{M}\left(q\right)\nu$ is the kinetic energy stored in the mechanical subsystem of the WMR. The time derivative of $V_{e}$ along the trajectories of system in (7), (8), $\dot{\phi }=w$, (5), (11) and (12) is given as:$${\dot{V}}_{e}=V^{T}\left[I_{c}-I-\frac{1}{R_{c}}V\right]+I_{c}^{T}\left[-V+EU\right]+I^{T}\left[-RI-nk_{m}\nu +V\right]+\nu^{T}\left[-\bar{C}\left(q,\dot{q}\right)\nu -f\nu +nk_{m}I\right].$$

Notice that several terms cancel to obtain:(21)$$\begin{array}{ccc} {\dot{V}}_{e} & = & -\frac{1}{R_{c}}{∥V∥}^{2}-{R∥I∥}^{2}-f{∥\nu ∥}^{2}+EI_{c}^{T}U. \end{array}$$

We stress that these term cancellations represent (1) energy exchange between the electrical and the mechanical subsystems of DC-motors, (2) energy exchange between the converter capacitor and the electrical subsystem of DC-motors, (3) energy exchange between the capacitor and the inductance of the Buck power electronic converters, and (4) skew-symmetry of matrix $\bar{C}\left(q,\dot{q}\right)$.

Hence, if we define the input EU and the output $I_{c}$, then the dynamical model in (7), (8), $\dot{\phi }=w$, (5), (11) and (12) is passive [37], [Ch. 6]. This property is exploited in this paper to design a path-tracking controller for the WMR when an inverter-DC/DC Buck power converter system is employed as power amplifier.

## 4. Main Result

Our main result is stated in the following proposition. The reader may find some similarities with controllers in [23,27]. The main reason for this is that the same inverter-DC/DC Buck power electronic converter system is considered but applied to very different plants. Thus, these similarities must be seen as indicatives that we found a powerful control methodology that is useful to control a set of different nonlinear plants. This is an important contribution of [23,27] and the present paper.

**Proposition** **1.**
*Consider the mathematical model in (7), (8), $\dot{\phi }=w$, (5), (11) and (12), in closed-loop with the following controller:*
(22)$$\begin{array}{ccc} U & = & \frac{1}{E}\left(V^{*}-K_{pc}{\tilde{I}}_{c}-K_{ic}\int_{0}^{t}{\tilde{I}}_{c}dr\right), \end{array}$$
(23)$$\begin{array}{ccc} I_{c}^{*} & = & \frac{1}{R_{c}}V^{*}-K_{pV}\tilde{V}+I^{*}-K_{iV}\int_{0}^{t}\tilde{V}dr, \end{array}$$
(24)$$\begin{array}{ccc} I^{*} & = & \frac{1}{k_{m}}\left(K_{p}\tilde{\nu }+K_{i}\int_{0}^{t}\tilde{\nu }\left(r\right)dr\right), \end{array}$$
(25)$$\begin{array}{ccc} V^{*} & = & RI^{*}-\alpha_{p}\tilde{I}-\alpha_{i}\int_{0}^{t}\tilde{I}dt, \end{array}$$
(26)$$\begin{array}{ccc} \nu_{d} & = & A^{-1}\left[\begin{array}{c} v_{d}\\ w_{d} \end{array}\right],\;\tilde{\nu }=\nu_{d}-\nu =\left[\begin{array}{c} {\tilde{\nu }}_{1}\\ {\tilde{\nu }}_{2} \end{array}\right], \end{array}$$
(27)$$\begin{array}{ccc} w_{d} & = & -k_{p\phi }\phi_{e}-k_{d\phi }\frac{d\phi_{e}}{dt}+k_{d\phi }w,\;\;\phi_{e}=\phi -\phi^{*}, \end{array}$$
*where ${\tilde{I}}_{c}=I_{c}-I_{c}^{*}$, $\tilde{V}=V-V^{*}$, $\tilde{I}=I-I^{*}$, $v_{d}>0$ is a constant standing for the WMR desired translational velocity and $K_{p}=\mathrm{diag}\left\{k_{p_{1}},k_{p_{2}}\right\}$, $K_{i}=\mathrm{diag}\left\{k_{i_{1}},k_{i_{2}}\right\}$. There always exist positive controller gains $k_{p\phi }$, $k_{d\phi }$, $k_{p_{1}}$, $k_{p_{2}}$, $k_{i_{1}}$, $k_{i_{2}}$ and diagonal positive definite matrices $\alpha_{p}$, $\alpha_{i}$, $K_{pc}$, $K_{ic}$, $K_{pV}$, $K_{iV}$, such that the closed-loop state is bounded and it has an ultimate bound which depends on the WMR translational velocity.*


Block diagram of control scheme in Proposition 1 is presented in Figure 4. It is composed by five main loops: (1) a PI controller for electric current through the inductor of the DC/DC Buck power electronic converter, (2) a PI controller for voltage at the DC/DC Buck power electronic converter output (at the capacitor terminals), (3) a PI controller for electric currents through the motor armatures, (4) a PI controller for motors velocity, and (5) a PD controller for path-tracking. Thus, our proposal contains the fundamental components in industrial applications and, hence, it is expected to be robust with respect to parametric uncertainties and external disturbances.

### 4.1. Closed-Loop Dynamics

Adding and subtracting some convenient terms in (7) and (8) and replacing (24), (25), we find: (28)$$\begin{array}{cc} \bar{M}\left(q\right)\dot{\tilde{\nu }}+\bar{C}\left(q,\dot{q}\right)\tilde{\nu }+f\tilde{\nu }= & -nk_{m}\tilde{I}+\bar{M}\left(q\right){\dot{\nu }}_{d}+\bar{C}\left(q,\dot{q}\right)\nu_{d}+f\nu_{d}-{\bar{K}}_{p}\tilde{\nu }-{\bar{K}}_{i}z, \end{array}$$(29)$$\begin{array}{cc} L_{a}\dot{\tilde{I}}= & -\left(R+\alpha_{p}\right)\tilde{I}+nk_{m}\tilde{\nu }+\tilde{V}-nk_{m}\nu_{d}-\alpha_{i}z_{I}-L_{a}{\dot{I}}^{*}, \end{array}$$(30)$$\begin{array}{cc} z= & \int_{0}^{t}\tilde{\nu }\left(r\right)dr,\;z_{I}=\int_{0}^{t}\tilde{I}dt, \end{array}$$
where we defined ${\bar{K}}_{p}=nK_{p}$, ${\bar{K}}_{i}=nK_{i}$. The reader may simply cancel all of the redundant terms in the above expressions to find the convenient terms that are added and subtracted. This applies in several parts in the present proof. Please note that according to (5) and (26), we have that:(31)$$\begin{array}{c} \tilde{\nu }=A^{-1}\left[\begin{array}{c} \tilde{v}\\ \tilde{w} \end{array}\right],\;\tilde{v}=v_{d}-v,\;\tilde{w}=w_{d}-w. \end{array}$$

Using $\dot{\phi }=w$ we can write: (32)$$\begin{array}{ccc} \dot{\phi } & = & w_{d}-\tilde{w}, \end{array}$$(33)$$\begin{array}{ccc} {\dot{\phi }}^{*} & = & w-\frac{d\phi_{e}}{dt}. \end{array}$$

Replacing (33) in $w_{d}$ given in (27) and replacing $w_{d}$ in (32), we obtain:(34)$${\dot{\phi }}_{e}=-\tilde{w}-k_{p\phi }\phi_{e}+\left(k_{d\phi }-1\right){\dot{\phi }}^{*}.$$

Adding and subtracting some convenient terms in (Equation 11), (12) and replacing (Equation 22), (23), we find: (35)$$\begin{array}{cc} L{\dot{\tilde{I}}}_{c}= & -\tilde{V}-K_{pc}{\tilde{I}}_{c}-K_{ic}z_{c}-L{\dot{I}}_{c}^{*}, \end{array}$$(36)$$\begin{array}{cc} C\dot{\tilde{V}}= & {\tilde{I}}_{c}-\tilde{I}-\left(\frac{1}{R_{c}}+K_{pV}\right)\tilde{V}-K_{iV}z_{V}-C{\dot{V}}^{*}, \end{array}$$(37)$$\begin{array}{cc} z_{c}= & \int_{0}^{t}{\tilde{I}}_{c}dr,\;z_{V}=\int_{0}^{t}\tilde{V}dr. \end{array}$$

The closed-loop dynamics is given by (Equation 28)–(30) and (Equation 34)–(37), and the state is:$$\zeta ={\left[{\tilde{\nu }}^{T},{\tilde{I}}^{T},{\tilde{V}}^{T},{\tilde{I}}_{c}^{T},z^{T},z_{I}^{T},z_{c}^{T},z_{V}^{T},\phi_{e}\right]}^{T}\in R^{17}.$$

### 4.2. Stability Analysis

We propose the following “energy” storage function for the closed-loop dynamics (see (Equation 20)):(38)$$\begin{array}{ccc} W(\zeta ) & = & \frac{C}{2}∥\tilde{V}{∥}^{2}+\frac{L}{2}∥{\tilde{I}}_{c}{∥}^{2}+\frac{1}{2}L_{a}∥\tilde{I}{∥}^{2}+\frac{1}{2}\alpha_{i}∥z_{I}{∥}^{2}+\frac{1}{2}K_{iV}∥z_{V}{∥}^{2}+\frac{1}{2}K_{ic}∥z_{c}{∥}^{2}+\frac{1}{2}\beta_{2}K_{pc}{∥z_{c}∥}^{2}\\ & + & \frac{1}{2}\beta_{1}\left(\frac{1}{R_{c}}+K_{pV}\right)∥z_{V}{∥}^{2}+\frac{1}{2}\beta_{3}\left(R+\alpha_{p}\right){∥z_{I}∥}^{2}+\beta_{1}C{\tilde{V}}^{T}z_{V}+\beta_{2}L{\tilde{I}}_{c}^{T}z_{c}+\beta_{3}L_{a}{\tilde{I}}^{T}z_{I}\\ & + & V_{\phi }\left(\phi_{e},\tilde{\nu },z\right), \end{array}$$
where:(39)$$\begin{array}{ccc} V_{\phi }\left(\phi_{e},\tilde{\nu },z\right) & = & \frac{1}{2}\phi_{e}^{2}+\frac{1}{2}{\tilde{\nu }}^{T}\bar{M}\left(q\right)\tilde{\nu }+z^{T}\bar{M}\left(q\right)\tilde{\nu }+\frac{1}{2}z^{T}{\bar{K}}_{i}z+\frac{1}{2}z^{T}{\bar{K}}_{p}z. \end{array}$$

Notice that (Equation 39) can be bounded as follows [33]:(40)$$\begin{array}{ccc} V_{\phi } & \geq & \frac{1}{2}{\left|\phi \right|}^{2}+\frac{1}{2}\lambda_{\min}\left\{\bar{M}\left(q\right)\right\}∥\tilde{\nu }{∥}^{2}-\lambda_{\max}\left\{\bar{M}\left(q\right)\right\}∥z∥∥\tilde{\nu }∥+\frac{1}{2}\lambda_{\min}\left\{{\bar{K}}_{i}\right\}{∥z∥}^{2}+\frac{1}{2}\lambda_{\min}\left\{{\bar{K}}_{p}\right\}{∥z∥}^{2},\\ & = & \frac{1}{2}{\left|\phi_{e}\right|}^{2}+\frac{1}{2}{\left[\begin{array}{c} ∥\tilde{\nu }∥\\ ∥z∥ \end{array}\right]}^{T}Q_{1}\left[\begin{array}{c} ∥\tilde{\nu }∥\\ ∥z∥ \end{array}\right],\\ & \geq & \frac{1}{2}|\phi_{e}{|}^{2}+\frac{1}{2}\lambda_{\min}\left\{Q_{1}\right\}[∥\tilde{\nu }{∥}^{2}+{∥z∥}^{2}]\geq \frac{1}{2}\min\left\{1,\lambda_{\min}\left\{Q_{1}\right\}\right\}{∥\xi ∥}^{2}, \end{array}$$
$$\begin{array}{ccc} Q_{1} & = & \left[\begin{array}{cc} \lambda_{\min}\left\{\bar{M}\left(q\right)\right\} & -\lambda_{\max}\left\{\bar{M}\left(q\right)\right\}\\ -\lambda_{\max}\left\{\bar{M}\left(q\right)\right\} & \lambda_{\min}\left\{{\bar{K}}_{i}\right\}+\lambda_{\min}\left\{{\bar{K}}_{p}\right\} \end{array}\right], \end{array}$$
with $\xi ={\left[\phi_{e},{\tilde{\nu }}_{1},{\tilde{\nu }}_{2},z_{1},z_{2}\right]}^{T}$. On the other hand, it is possible to upper bound (Equation 39) as [33]:(41)$$\begin{array}{ccc} V_{\phi } & \leq & \frac{1}{2}{\left|\phi \right|}^{2}+\frac{1}{2}\lambda_{\max}\left\{\bar{M}\left(q\right)\right\}∥\tilde{\nu }{∥}^{2}+\lambda_{\max}\left\{\bar{M}\left(q\right)\right\}∥z∥∥\tilde{\nu }∥+\frac{1}{2}\lambda_{\max}\left\{{\bar{K}}_{i}\right\}{∥z∥}^{2}+\frac{1}{2}\lambda_{\max}\left\{{\bar{K}}_{p}\right\}{∥z∥}^{2},\\ & = & \frac{1}{2}{\left|\phi_{e}\right|}^{2}+\frac{1}{2}{\left[\begin{array}{c} ∥\tilde{\nu }∥\\ ∥z∥ \end{array}\right]}^{T}Q_{2}\left[\begin{array}{c} ∥\tilde{\nu }∥\\ ∥z∥ \end{array}\right],\\ & \leq & \frac{1}{2}|\phi_{e}{|}^{2}+\frac{1}{2}\lambda_{\max}\left\{Q_{2}\right\}[∥\tilde{\nu }{∥}^{2}+{∥z∥}^{2}]\leq \frac{1}{2}\max\left\{1,\lambda_{\max}\left\{Q_{2}\right\}\right\}{∥\xi ∥}^{2}, \end{array}$$
$$\begin{array}{ccc} Q_{2} & = & \left[\begin{array}{cc} \lambda_{\max}\left\{\bar{M}\left(q\right)\right\} & \lambda_{\max}\left\{\bar{M}\left(q\right)\right\}\\ \lambda_{\max}\left\{\bar{M}\left(q\right)\right\} & \lambda_{\max}\left\{{\bar{K}}_{i}\right\}+\lambda_{\max}\left\{{\bar{K}}_{p}\right\} \end{array}\right]. \end{array}$$

Moreover, we can write:$$\begin{array}{ccc} & & \frac{1}{2}\chi_{1}^{T}P_{1}\chi_{1}\leq \frac{C}{2}∥\tilde{V}{∥}^{2}+\beta_{1}C{\tilde{V}}^{T}z_{V}+\frac{1}{2}K_{iV}{∥z_{V}∥}^{2}\leq \frac{1}{2}\chi_{1}^{T}{\bar{P}}_{1}\chi_{1},\\ & & \frac{1}{2}\chi_{2}^{T}P_{2}\chi_{2}\leq \frac{L}{2}∥{\tilde{I}}_{c}{∥}^{2}+\beta_{2}L{\tilde{I}}_{c}^{T}z_{c}+\frac{1}{2}K_{ic}{∥z_{c}∥}^{2}\leq \frac{1}{2}\chi_{2}^{T}{\bar{P}}_{2}\chi_{2},\\ & & \frac{1}{2}\chi_{3}^{T}P_{3}\chi_{3}\leq \frac{1}{2}L_{a}∥\tilde{I}{∥}^{2}+\beta_{3}L_{a}{\tilde{I}}^{T}z_{I}+\frac{1}{2}\alpha_{i}{∥z_{I}∥}^{2}\leq \frac{1}{2}\chi_{3}^{T}{\bar{P}}_{3}\chi_{3}, \end{array}$$
$$\begin{array}{ccc} P_{1} & = & \left[\begin{array}{cc} C & -\beta_{1}C\\ -\beta_{1}C & K_{iV} \end{array}\right],\;{\bar{P}}_{1}=\left[\begin{array}{cc} C & \beta_{1}C\\ \beta_{1}C & K_{iV} \end{array}\right],\\ P_{2} & = & \left[\begin{array}{cc} L & -\beta_{2}L\\ -\beta_{2}L & K_{ic} \end{array}\right],\;{\bar{P}}_{2}=\left[\begin{array}{cc} L & \beta_{2}L\\ \beta_{2}L & K_{ic} \end{array}\right],\\ P_{3} & = & \left[\begin{array}{cc} L_{a} & -\beta_{3}L_{a}\\ -\beta_{3}L_{a} & \alpha_{i} \end{array}\right],\;{\bar{P}}_{3}=\left[\begin{array}{cc} L_{a} & \beta_{3}L_{a}\\ \beta_{3}L_{a} & \alpha_{i} \end{array}\right], \end{array}$$
where we defined $\chi_{1}=[∥\tilde{V}∥,∥z_{V}{∥]}^{T}$, $\chi_{2}=[∥{\tilde{I}}_{c}∥,∥z_{c}{∥]}^{T}$, and $\chi_{3}=[∥\tilde{I}∥,∥z_{I}{∥]}^{T}$. Thus, according to this and (Equation 40) and (Equation 41), we have that
(42)$$\begin{array}{c} \alpha_{1}\left(∥\zeta ∥\right)=c_{1}{∥\zeta ∥}^{2}\leq W\left(\zeta \right)\leq c_{2}{∥\zeta ∥}^{2}=\alpha_{2}\left(∥\zeta ∥\right), \end{array}$$
where:$$\begin{array}{c} 0<c_{1}=\frac{1}{2}\min\{1,\lambda_{\min}\left\{Q_{1}\right\},\lambda_{\min}\left\{P_{1}\right\},\lambda_{\min}\left\{P_{2}\right\},\\ \lambda_{\min}\left\{P_{3}\right\},\frac{1}{2}\beta_{2}K_{pc},\frac{1}{2}\beta_{1}\left(\frac{1}{R_{c}}+K_{pV}\right),\frac{1}{2}\beta_{3}\left(R+\alpha_{p}\right)\},\\ c_{2}=\frac{1}{2}\max\{1,\lambda_{\max}\left\{Q_{2}\right\},\lambda_{\max}\left\{{\bar{P}}_{1}\right\},\lambda_{\max}\left\{{\bar{P}}_{2}\right\},\\ \lambda_{\max}\left\{{\bar{P}}_{3}\right\},\frac{1}{2}\beta_{2}K_{pc},\frac{1}{2}\beta_{1}\left(\frac{1}{R_{c}}+K_{pV}\right),\frac{1}{2}\beta_{3}\left(R+\alpha_{p}\right)\}. \end{array}$$

Notice that $\alpha_{1}$ and $\alpha_{2}$ are class $K_{\infty }$ functions.

Taking advantage from the following cancellations, which are a direct consequence of the similarities between the closed-loop dynamics (Equation 28)–(30) and (Equation 34)–(37), and the open-loop dynamics (7), (8), $\dot{\phi }=w$, (5), (Equation 11) and (12), (also see the paragraph after (Equation 21)):$$\begin{array}{c} {\tilde{V}}^{T}{\tilde{I}}_{c}-{\tilde{I}}_{c}^{T}\tilde{V}=0,\;-{\tilde{V}}^{T}\tilde{I}+{\tilde{I}}^{T}\tilde{V}=0,\;{\tilde{\nu }}^{T}\bar{C}\left(q,\dot{q}\right)\tilde{\nu }=0, \end{array}$$
we find that the time derivative of *W* along the trajectories of the closed-loop system (Equation 28)–(30) and (Equation 34)–(37), is given as:$$\begin{array}{ccc} \dot{W} & = & {\tilde{V}}^{T}\left[-\left(\frac{1}{R_{c}}+K_{pV}\right)\tilde{V}-K_{iV}z_{V}-C{\dot{V}}^{*}\right]+{\tilde{I}}_{c}^{T}\left[-K_{pc}{\tilde{I}}_{c}-K_{ic}z_{c}-L{\dot{I}}_{c}^{*}\right]\\ & + & {\tilde{I}}^{T}\left[-\left(R+\alpha_{p}\right)\tilde{I}+nk_{m}\tilde{\nu }-nk_{m}\nu_{d}-\alpha_{i}z_{I}-L_{a}{\dot{I}}^{*}\right]+\alpha_{i}z_{I}^{T}\tilde{I}+K_{iV}z_{V}^{T}\tilde{V}+K_{ic}z_{c}^{T}{\tilde{I}}_{c}+\beta_{1}C{\tilde{V}}^{T}\tilde{V}\\ & + & \beta_{1}z_{V}^{T}\left[{\tilde{I}}_{c}-\tilde{I}-K_{iV}z_{V}-C{\dot{V}}^{*}\right]+\beta_{2}L{\tilde{I}}_{c}^{T}{\tilde{I}}_{c}+\beta_{2}z_{c}^{T}\left[-\tilde{V}-K_{ic}z_{c}-L{\dot{I}}_{c}^{*}\right]+\beta_{3}L_{a}{\tilde{I}}^{T}\tilde{I}\\ & + & \beta_{3}z_{I}^{T}\left[nk_{m}\tilde{\nu }+\tilde{V}-nk_{m}\nu_{d}-\alpha_{i}z_{I}-L_{a}{\dot{I}}^{*}\right]+{\dot{V}}_{\phi }, \end{array}$$
where:$$\begin{array}{ccc} {\dot{V}}_{\phi }= & - & \tilde{w}\phi_{e}-k_{p_{\phi }}\phi_{e}^{2}+\phi_{e}\left(k_{d\phi }-1\right){\dot{\phi }}^{*}-\frac{1}{2}{\tilde{\nu }}^{T}{\bar{K}}_{p}\tilde{\nu }-\frac{1}{2}{\tilde{\nu }}^{T}\left[{\bar{K}}_{p}-2\bar{M}\left(q\right)\right]\tilde{\nu }-z^{T}\bar{C}\left(q,\dot{q}\right)\tilde{\nu }-z^{T}{\bar{K}}_{i}z\\ & + & {\left(\tilde{\nu }+z\right)}^{T}\left[\bar{M}\left(q\right){\dot{\nu }}_{d}+\bar{C}\left(q,\dot{q}\right)\nu_{d}+f\nu_{d}\right]-{\left(\tilde{\nu }+z\right)}^{T}nk_{m}\tilde{I}-{\left(\tilde{\nu }+z\right)}^{T}f\tilde{\nu }. \end{array}$$

We can bound the following term as [33]:(43)$$\begin{array}{ccc} & & -\frac{1}{2}z^{T}{\bar{K}}_{i}z+z^{T}\left[\bar{M}\left(q\right){\dot{\nu }}_{d}+\bar{C}\left(q,\dot{q}\right)\nu_{d}+f\nu_{d}\right]\leq -\frac{1}{2}\lambda_{\min}\left\{{\bar{K}}_{i}\right\}{∥z∥}^{2}\\ & & +∥z∥\left[\lambda_{\max}\left\{\bar{M}\left(q\right)\right\}∥{\dot{\nu }}_{d}∥+\frac{r^{2}}{2b}m_{c}d|\dot{\phi }|∥\nu_{d}∥+f∥\nu_{d}∥\right],\\ & & \leq -\frac{1}{2}\lambda_{\min}\left\{{\bar{K}}_{i}\right\}{∥z∥}^{2}+∥z∥s_{3}\leq 0,\;\mathrm{if}\;\frac{2s_{3}}{\lambda_{\min}\left\{{\bar{K}}_{i}\right\}}\leq ∥z∥\leq ∥x∥,\\ s_{3} & = & ∥A^{-1}∥[\lambda_{\max}\left\{\bar{M}\left(q\right)\right\}\sqrt{\max{\left\{k_{p\phi },k_{d\phi }\right\}}^{2}\left(\right|{\dot{\phi }}_{e}\left|+\right|{\ddot{\phi }}_{e}\left|+\right|\ddot{\phi }{\left|\right)}^{2}+{\dot{v}}_{d}^{2}}\\ & & +\frac{r^{2}}{2b}m_{c}d\left|\dot{\phi }\right|\sqrt{\max{\left\{k_{p\phi },k_{d\phi }\right\}}^{2}\left(\right|\phi_{e}\left|+\right|{\dot{\phi }}_{e}\left|+\right|\dot{\phi }{\left|\right)}^{2}+v_{d}^{2}}\\ & & +f\sqrt{\max{\left\{k_{p\phi },k_{d\phi }\right\}}^{2}\left(\right|\phi_{e}\left|+\right|{\dot{\phi }}_{e}\left|+\right|\dot{\phi }{\left|\right)}^{2}+v_{d}^{2}}]. \end{array}$$

In a similar way, we can obtain:(44)$$\begin{array}{ccc} & & -\frac{1}{2}{\tilde{\nu }}^{T}{\bar{K}}_{p}\tilde{\nu }+{\tilde{\nu }}^{T}\left[\bar{M}\left(q\right){\dot{\nu }}_{d}+\bar{C}\left(q,\dot{q}\right)\nu_{d}+f\nu_{d}\right]\leq 0,\;\mathrm{if}\;\frac{2s_{3}}{\lambda_{\min}\left\{{\bar{K}}_{p}\right\}}\leq ∥\tilde{\nu }∥\leq ∥x∥. \end{array}$$

Therefore, we can bound ${\dot{V}}_{\phi }$ as:$$\begin{array}{ccc} {\dot{V}}_{\phi }\leq & - & k_{p\phi }{\left|\phi_{e}\right|}^{2}-\frac{1}{2}{\tilde{\nu }}^{T}\underset{K_{PM}}{\underbrace{\left[{\bar{K}}_{p}-2\bar{M}\left(q\right)\right]}}\tilde{\nu }-\frac{1}{2}z^{T}{\bar{K}}_{i}z\\ & - & \tilde{w}\phi_{e}+\left(k_{d\phi }-1\right)\phi_{e}{\dot{\phi }}^{*}-z^{T}\bar{C}\left(q,\dot{q}\right)\tilde{\nu }-{\left(\tilde{\nu }+z\right)}^{T}nk_{m}\tilde{I}-{\left(\tilde{\nu }+z\right)}^{T}f\tilde{\nu }. \end{array}$$

Matrix $K_{PM}$ is positive definite (and, hence, all of its eigenvalues are positive) if it is strictly diagonally dominant and its diagonal entries are positive [38], i.e.,:$$\begin{array}{ccc} nk_{p_{1}}-2(|{\bar{M}}_{11}\left|+\right|{\bar{M}}_{12}\left|\right) & > & 0,\\ nk_{p_{2}}-2(|{\bar{M}}_{21}\left|+\right|{\bar{M}}_{22}\left|\right) & > & 0. \end{array}$$

Considering (Equation 3), (Equation 31) and using matrix notation, we can write [33]:(45)$$\begin{array}{c} {\dot{V}}_{\phi }\leq -u^{T}Q_{0}u+|\phi_{e}\left|\right|k_{d_{\phi }}-1||{\dot{\phi }}^{*}|-{\left(\tilde{\nu }+z\right)}^{T}nk_{m}\tilde{I}, \end{array}$$
and matrix $Q_{0}$ is defined as:$$\left[\begin{array}{ccccc} k_{p\phi } & -\frac{r}{4b} & -\frac{r}{4b} & 0 & 0\\ -\frac{r}{4b} & \frac{1}{2}\lambda_{1}\left\{K_{PM}\right\}+f & 0 & -\frac{f}{2} & \;-\frac{r^{2}}{4}m_{c}d\left|\dot{\phi }\right|\\ -\frac{r}{4b} & 0 & \;\frac{1}{2}\lambda_{2}\left\{K_{PM}\right\}+f & -\frac{r^{2}}{4}m_{c}d\left|\dot{\phi }\right| & -\frac{f}{2}\\ 0 & -\frac{f}{2} & -\frac{r^{2}}{4}m_{c}d\left|\dot{\phi }\right| & \frac{1}{2}nk_{i_{1}} & 0\\ 0 & -\frac{r^{2}}{4}m_{c}d\left|\dot{\phi }\right| & -\frac{f}{2} & 0 & \frac{1}{2}nk_{i_{2}} \end{array}\right]$$
where $u=\left[\right|\phi_{e}\left|,\right|{\tilde{\nu }}_{1}\left|,\right|\tilde{\nu_{2}}\left|,\right|z_{1}\left|,\right|z_{2}{\left|\right]}^{T}$ and $\lambda_{1}\left\{K_{PM}\right\}$, $\lambda_{2}\left\{K_{PM}\right\}$, represent two positive constants such that $0<\lambda_{1}\left\{K_{PM}\right\}<\lambda_{\min}\left\{K_{PM}\right\}$ and $0<\lambda_{2}\left\{K_{PM}\right\}<\lambda_{\min}\left\{K_{PM}\right\}$.

On the other hand, taking into account $nk_{m}{\tilde{I}}^{T}\tilde{\nu }-nk_{m}{\tilde{\nu }}^{T}\tilde{I}=0$, and (Equation 45), it is found that $\dot{W}$ can be upper bounded as:(46)$$\begin{array}{ccc} \dot{W} & \leq & -\left(\frac{1}{R_{c}}+K_{pV}-\beta_{1}C\right){\tilde{V}}^{T}\tilde{V}-\left(K_{pc}-\beta_{2}L\right){\tilde{I}}_{c}^{T}{\tilde{I}}_{c}-\left(R+\alpha_{p}-\beta_{3}L_{a}\right){\tilde{I}}^{T}\tilde{I}-\beta_{1}K_{iV}z_{V}^{T}z_{V}\\ & & -\beta_{2}K_{ic}z_{c}^{T}z_{c}-\beta_{3}\alpha_{i}z_{I}^{T}z_{I}-{\left(\tilde{V}+\beta_{1}z_{V}\right)}^{T}C{\dot{V}}^{*}-{\left({\tilde{I}}_{c}+\beta_{2}z_{c}\right)}^{T}L{\dot{I}}_{c}^{*}-{\left(\tilde{I}+\beta_{3}z_{I}\right)}^{T}L_{a}{\dot{I}}^{*}\\ & & -nk_{m}{\tilde{I}}^{T}\nu_{d}+\beta_{1}z_{V}^{T}\left[{\tilde{I}}_{c}-\tilde{I}\right]-\beta_{2}z_{c}^{T}\tilde{V}+\beta_{3}z_{I}^{T}\left[nk_{m}\tilde{\nu }+\tilde{V}-nk_{m}\nu_{d}\right]\\ & & -u^{T}Q_{0}u+|\phi_{e}\left|\right|k_{d_{\phi }}-1||{\dot{\phi }}^{*}|-z^{T}nk_{m}\tilde{I}. \end{array}$$

Moreover, recalling (23), (24) and (25), we have that:$$\begin{array}{ccc} {\dot{V}}^{*} & = & \frac{R}{k_{m}}\left(K_{p}{\bar{M}}^{-1}\left(q\right)\left[-\bar{C}\left(q,\dot{q}\right)\tilde{\nu }-nk_{m}\tilde{I}+\bar{M}\left(q\right){\dot{\nu }}_{d}+\bar{C}\left(q,\dot{q}\right)\nu_{d}-{\bar{K}}_{p}\tilde{\nu }-{\bar{K}}_{i}z\right]+K_{i}\tilde{\nu }\right)\\ & - & \frac{\alpha_{p}}{L_{a}}\{-\left(R+\alpha_{p}\right)\tilde{I}+nk_{m}\tilde{\nu }+\tilde{V}-nk_{m}\nu_{d}-\alpha_{i}z_{I}-\frac{L_{a}}{k_{m}}(K_{p}{\bar{M}}^{-1}\left(q\right)[-\bar{C}\left(q,\dot{q}\right)\tilde{\nu }-nk_{m}\tilde{I}\\ & + & \bar{M}\left(q\right){\dot{\nu }}_{d}+\bar{C}\left(q,\dot{q}\right)\nu_{d}-{\bar{K}}_{p}\tilde{\nu }-{\bar{K}}_{i}z]+K_{i}\tilde{\nu })\}-\alpha_{i}\tilde{I},\\ {\dot{I}}_{c}^{*} & = & \left(\frac{1}{R_{c}}+K_{pV}\right){\dot{V}}^{*}-\frac{K_{pV}}{C}\left[{\tilde{I}}_{c}-\tilde{I}-\left(\frac{1}{R_{c}}+K_{pV}\right)\tilde{V}-K_{iV}z_{V}\right]\\ & + & \frac{1}{k_{m}}\left(K_{p}{\bar{M}}^{-1}\left(q\right)\left[-\bar{C}\left(q,\dot{q}\right)\tilde{\nu }-nk_{m}\tilde{I}+\bar{M}\left(q\right){\dot{\nu }}_{d}+\bar{C}\left(q,\dot{q}\right)\nu_{d}-{\bar{K}}_{p}\tilde{\nu }-{\bar{K}}_{i}z\right]+K_{i}\tilde{\nu }\right)-K_{iV}\tilde{V},\\ {\dot{I}}^{*} & = & \frac{1}{k_{m}}\left(K_{p}{\bar{M}}^{-1}\left(q\right)\left[-\bar{C}\left(q,\dot{q}\right)\tilde{\nu }-nk_{m}\tilde{I}+\bar{M}\left(q\right){\dot{\nu }}_{d}+\bar{C}\left(q,\dot{q}\right)\nu_{d}-{\bar{K}}_{p}\tilde{\nu }-{\bar{K}}_{i}z\right]+K_{i}\tilde{\nu }\right). \end{array}$$

Thus, we can write (Equation 46) as:(47)$$\begin{array}{ccc} \dot{W} & \leq & -\rho^{T}P\rho +∥\rho ∥\mu_{0}, \end{array}$$
where:$$\begin{array}{c} \rho =\left[\right|\phi_{e}\left|,\right|{\tilde{\nu }}_{1}\left|,\right|\tilde{\nu_{2}}\left|,\right|z_{1}\left|,\right|z_{2}|,∥\tilde{I}∥,∥z_{I}∥,∥\tilde{V}∥,∥z_{V}∥,∥{\tilde{I}}_{c}∥,∥z_{c}{∥]}^{T}, \end{array}$$
and $\mu_{0}>0$ is proportional to either $s_{3}$ or $|{\dot{\phi }}^{*}|$. Entries of matrix *P* are given as follows.

The first five rows and columns of matrix *P* are identical to $Q_{0}$ defined after (Equation 45).All entries of *P* are constant or depend on $|\dot{\phi }|$.All of the diagonal entries of *P* depend on some controller gain. This controller gain is different for each diagonal entry.Let $\gamma_{i}$ be the controller gain the $P_{ii}$ entry depends on. Excepting $P_{ii}$, any of the entries of the submatrix defining the i-th leading principal minor do not depend on $\gamma_{i}$.According to the previous item, given some $\sup_{t\geq 0}\left|\dot{\phi }\left(t\right)\right|$, there always exists a large enough $\gamma_{i}>0$ such that the i-th leading principal minor can be rendered positive.Given some $\sup_{t\geq 0}\left|\dot{\phi }\left(t\right)\right|$, there always exist controller gains such that matrix *P* is positive definite, i.e., $\lambda_{\min}\left\{P\right\}>0$.

Hence, recalling $-\rho^{T}P\rho \leq -\lambda_{\min}\left\{P\right\}{∥\rho ∥}^{2}$ and given some 0<Θ<1, we can write (Equation 47) as:(48)$$\begin{array}{ccc} \dot{W} & \leq & -\left(1-\Theta \right)\lambda_{\min}{\left\{P\right\}∥\rho ∥}^{2}-\Theta \lambda_{\min}{\left\{P\right\}∥\rho ∥}^{2}+∥\rho ∥\mu_{0},\\ \dot{W} & \leq & -\left(1-\Theta \right)\lambda_{\min}{\left\{P\right\}∥\rho ∥}^{2},\;\mathrm{if}\;-\Theta \lambda_{\min}{\left\{P\right\}∥\rho ∥}^{2}+∥\rho ∥\mu_{0}<0,\\ \dot{W} & \leq & -\left(1-\Theta \right)\lambda_{\min}{\left\{P\right\}∥\rho ∥}^{2},\;\forall \;∥\rho ∥>\frac{\mu_{0}}{\Theta \lambda_{\min}\left\{P\right\}}=\sigma_{0}>0. \end{array}$$

Thus, according to Theorem 1, the ultimate bound is given as $\sqrt{\frac{c_{2}}{c_{1}}}\sigma_{0}$. Since $\lambda_{\min}\left\{P\right\}$ cannot be rendered arbitrary large by suitable selection of controller gains, this ultimate bound cannot be rendered arbitrarily small and we depend on a small enough $\mu_{0}$, i.e., on small enough $s_{3}$ and $|{\dot{\phi }}^{*}|$.

Thus, invoking Theorem 1, it is proven that $\zeta \in R^{17}$ is bounded and it has an ultimate bound that decreases to zero as $s_{3}$ and $|{\dot{\phi }}^{*}|$ decrease to zero, i.e., as the WMR translational velocity *v* decreases to zero. Notice that $\phi_{e}$, $\dot{\phi }$, ${\dot{\phi }}^{*}$ and their time derivatives are larger as WMR moves faster, i.e., as *v* is larger. However, neither $s_{3}$ nor $|{\dot{\phi }}^{*}|$ can be zero since this would imply that WMR does not move. Thus, the ultimate bound cannot be zero. This completes the proof of Proposition 1. Conditions for this stability result are summarized by $c_{1}>0$ and all of the eleven leading principal minors of matrix *P* defined in (Equation 47) are positive.

**Remark** **1.**
*According to arguments at the end of the above proof, given any $|\dot{\phi }|$ there always exist controller gains such that $\lambda_{\min}\left\{P\right\}>0$ globally. However, large values of $|\dot{\phi }|$ also result in a large ultimate bound $\sqrt{\frac{c_{2}}{c_{1}}}\sigma_{0}$. This and the fact that sensors used to measure $\phi_{e}$ always have a limited range, force the stability result in Proposition 1 to be local. Moreover, according to Assumption 1, in Section 2.2, we require tracking error $\phi_{e}$ to be small in order to ensure that path-tracking is achieved.*


**Remark** **2.**
*As stated in the introduction Section, the most common control scheme for path tracking in industrial applications is composed by an external PID or PD controller for path tracking and an internal PI velocity control for both WMR wheels. In this respect, work in [33] constitutes the only result presenting a formal stability analysis for such a control scheme, i.e., by considering both the kinematic and dynamic models of WMR. We stress, again, that result in Proposition 1 represents an extension of result in [33] to the case when both (i) electrical dynamics of DC-motors used as actuators and (ii) dynamics of the inverter-DC/DC Buck power electronic converters used as power amplifiers, are taken into account during the stability analysis. Moreover, dynamics of all the above cited components are considered together in the stability analysis. This is different from some results in the literature [19], on DC-motor control, where hierarchical arguments are employed to assume that dynamics of the subsystems are isolated.*

*It is important to point out that because of the fast dynamics of both the electric dynamics of DC-motors and dynamics of the inverter-DC/DC Buck power electronic converters, no trajectory tracking performance differences are expected between control schemes in Proposition 1 and [33]. The reason to include inverter-DC/DC Buck power electronic converters as power amplifiers is to avoid stress of the DC-motors produced by the hard commutation that is intrinsic to PWM-based power amplifiers [1]. Hence, performance improvement is expected to occur at the practical power electronic stage. However, verifying this affirmation concerns to power electronics practitioners who have proposed such a solution. The present paper is control oriented and, thus, we are only interested in finding conditions ensuring stability of the closed-loop systems; stability conditions that can be used as tuning guidelines in practice.*


**Remark** **3.**
*Proposition 1 only ensures ultimate boundedness instead of asymptotic stability. However, we stress that this result is consistent with current practice when classical PI controllers are employed. As a matter of fact, it is well known from classical control that the integral part of a PI controller is intended for regulation tasks, i.e., for constant references or constant external disturbances. In this respect, it is clear from (Equation 48) that $\sigma_{0}$ is a disturbance avoiding asymptotic stability. If this disturbance was constant, then the PI controllers could compensate it and asymptotic stability would be achieved. However, notice that $\sigma_{0}$ constant implies $s_{3}$ constant, i.e., $\phi_{e},{\dot{\phi }}_{e}{\ddot{\phi }}_{e}$, and $\dot{\phi },\ddot{\phi }$ constants. Hence, this condition would restrict the result to track ingvery particular paths such as straight lines, i.e., the result would be very limited.*

*Moreover, the reader might argue that path-tracking is performed by a PD controller, but not by a PI controller. However, a PD controller cannot achieve a zero tracking error if DC-motors do not generate the desired torques demanded by the PD path-tracking controller. This requires electric currents in motor armatures to reach their desired values. This is not possible with PI electric current controllers if the desired currents are not constant because the desired torques are no constant. Thus, perfect tracking cannot be achieved when employing internal loops driven by classical PI controllers.*

*We are interested in proving that simple to implement controllers can achieve the path-tracking task. After all, simple to implement controllers are preferred by practitioners. We know that there exist advanced controllers ensuring zero tracking error in theory. However, most of those controllers only consider the mechanical part of the system and the electrical dynamics is neglected. What most people forget is that the main argument used to disregard the electric dynamics is the assumption that internal electric current loops driven by classical PI controllers are employed. This explains why a nonzero trajectory tracking error is reported when experimentally testing some advanced trajectory tracking controllers. We invite the reader to consult [40], for instance, where such a problem appears but, incorrectly, the authors attribute the nonzero tracking error to a wrong estimation of velocity.*


## 5. Simulation Results

In this section, we present a numerical example to give some insight on the achievable performance with the control scheme in Proposition 1. To this aim we employ the numerical parameters of the WMR built at the CIDETEC-IPN Mechatronics Laboratory (also see [33]). This is a differentially driven robot which was designed to exactly meet the description presented in Figure 1.

The robot total mass is 20 [Kg] and 0.422 [m] length, 0.35 [m] width, 0.25 [m] height are its geometric dimensions. Robot parameters are: r=0.07499 [m], b=0.175 [m], $m_{c}=15.8$ [Kg], $m_{w}=2.47$ [Kg], $I_{c}=0.135$ [Kgm${}^{2}$], $I_{m}=0.0015$ [Kgm${}^{2}$], $I_{w}=0.00375$ [Kgm${}^{2}$], d=0.05 [m]. It is assumed that the path-tracking sensor consists of a set of discrete optical sensors placed on an arc of circle centered at $P_{0}$ with $L_{0}=0.237$ [m] as radius. See Figure 3. This arc of circle is centered at the longitudinal axis of WMR spanning a total angle of 0.5 [rad]. In the following simulation results, we assume that separation among the discrete optical sensors is 0.005 [rad].

The two identical permanent magnet brushed DC-motors employed to provide torque have as parameters $k_{m}=1.748$ [Nm/A], R=1.05 [Ohm], $L_{a}=1.4\times 10^{-3}$ [H], n=1, f=0.5 [Nm/(rad/s)]. Each DC-motor is fed by a inverter-DC/DC Buck power electronic converter with the parameters E=20 [V], $L=10.6\times 10^{-3}$ [H], $R_{c}=100$ [Ohm], $C=220\times 10^{-6}$ [F].

The controller gains were chosen as $k_{p\phi }=4$, $k_{d\phi }=0.3$, $K_{p}=\mathrm{diag}\left\{6,6\right\}$, $K_{i}=\mathrm{diag}\left\{6,6\right\}$, $\alpha_{p}=\mathrm{diag}\left\{0.5,0.5\right\}$, $\alpha_{i}=\mathrm{diag}\left\{12,12\right\}$, $K_{pV}=\mathrm{diag}\left\{2,2\right\}$, $K_{iV}=\mathrm{diag}\left\{15,15\right\}$, $K_{pc}=\mathrm{diag}\left\{5,5\right\}$, $K_{ic}=\mathrm{diag}\left\{20,20\right\}$.

The desired path is eight-shaped, generated as two circles one above the other, each one with a 1 [m] radius. The upper circle is centered at (x,y)=(0,−1) in meters and circle at the bottom is centered at (x,y)=(0,−3) in meters. See Figure 5a. The desired translational velocity was fixed to be $v_{d}=0.5$ [m/s]. All initial conditions were fixed to zero, excepting y(0)=0.1 [m].

The desired path and the actual path described by robot are shown in Figure 5a. Notice that convergence to the desired path is achieved despite WMR is not on the desired path at the beginning. We realize that the actual path and the desired path overlap most of the time. In Figure 5b we observe that the tracking error angle $\phi_{e}$ never stays at zero but its maximal value is about 0.1 [rad]. Its initial large values are because WMR is not on the path at the beginning. Notice that $\phi_{e}$ changes sign when the sign of curvature of desired path also changes. In Figure 5c we present the WMR orientation, ϕ, just to verify that it corresponds to the described path. In Figure 5d we realize that the translational velocity reaches its desired value $v_{d}=0.5$ [m/s] very fast and remains there.

In Figure 6a,b we observe that electric currents through both converter inductors $i_{c1},i_{c2}$, remain within a reasonable range, i.e., about [0,4] [A]. The noise content is due to the discrete nature of the optical sensors and because the nature of the desired path is also discrete, given the discrete nature of these simulations. This feature results in noisy signals because of the derivative part of the PD path-tracking controller. These results were obtained using increments of 0.001 [rad] when computing circles defining the desired path. In Figure 6c,d we observe that voltages applied to both motors $V_{1},V_{2}$, are also noisy but these variables take values within a reasonable range, i.e., [0,20] [Volts].

In Figure 7a,b we see that electric currents through the motors $I_{1},I_{2}$, are in the range [0,4] [A], which is acceptable for this WMR prototype. Finally, in Figure 7c,d we see the variables $u_{1},u_{2}$, used to control the inverter transistors. These variables contain many large spikes which, however, are applied during short intervals of time. We also see that disregarding these spikes, these noisy signals remain in the range [0,1.5]. Since these variables can only take values in the range [−1,1], we saturated the variables plotted in Figure 7c,d to the range [−1,1] before applying them to transistors. As we saw above, this suffices to accomplish the path-tracking task.

As stated in Remark 2, performance improvement with control scheme in Proposition 1 with respect to experimental results in [33] is expected to occur only at the practical power electronic stage. Thus, we refer the reader to [33] in order to observe the tracking performance achieved in experiments. The present work is not concerned with the practical power electronic stage but it is only interested in finding conditions ensuring stability of the closed-loop system. Moreover, comparison with any previous work would be unfair because, until now, our proposal is the only one solving the problem at hand. See Remark 2.

## 6. Conclusions

We presented, for the first time, a formal stability result for a wheeled mobile robot when the DC-motors used as actuators are fed by inverter-Buck DC/DC power electronic converter systems as power amplifiers. It is assumed that robot is performing the path-tacking control task. Although asymptotic stability is not ensured, the proven ultimate boundedness of the state is consistent with current practice. This is because several internal PI loops are employed, which ensure zero steady state error only in regulation tasks. We point out that a nonzero steady state tracking error is common in practice even when employing advanced trajectory tracking controllers for mechanical systems. This is because it is assumed during the design stage that electrical dynamics may be neglected because PI electric current controllers are employed in practice. Thus, presenting a formal justification for this control scheme that is widely employed in industrial practice is another important contribution. Thus contrary to modern control strategies, our control scheme is simple to implement and we hope this feature results in a good acceptance from the part of practitioners.

The electrical signals that we obtain in simulations are noisy. This is because of the discrete nature of the methodology employed to measure the tracking error in our discrete simulations. This is magnified by the PD controller that we use for path-tracking. This feature is not restricted to our control scheme: highly noisy electrical signals appear whenever a controller with derivative action is employed to control the mechanical variables. We showed that despite these noisy electrical variables, the path-tracking task is accomplished satisfactorily.

## Figures and Tables

**Figure 1 sensors-20-06522-f001:**
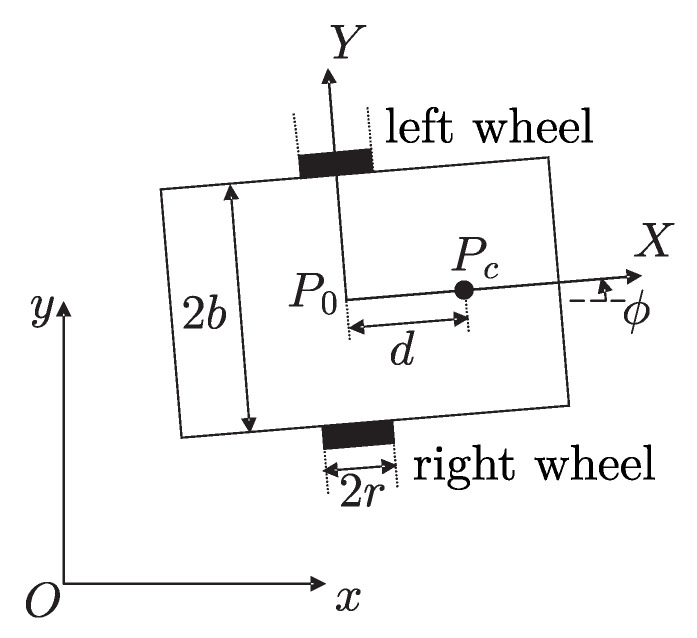
Wheeled mobile robot under study.

**Figure 2 sensors-20-06522-f002:**
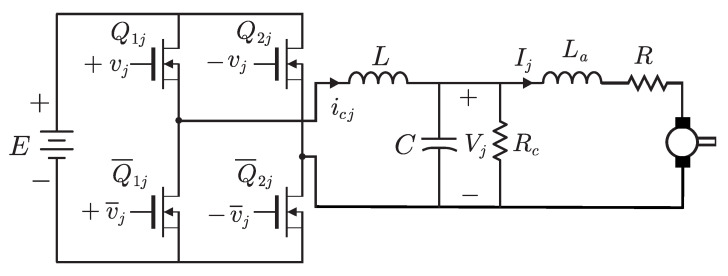
The inverter-DC/DC Buck power electronic converter-DC motor system.

**Figure 3 sensors-20-06522-f003:**
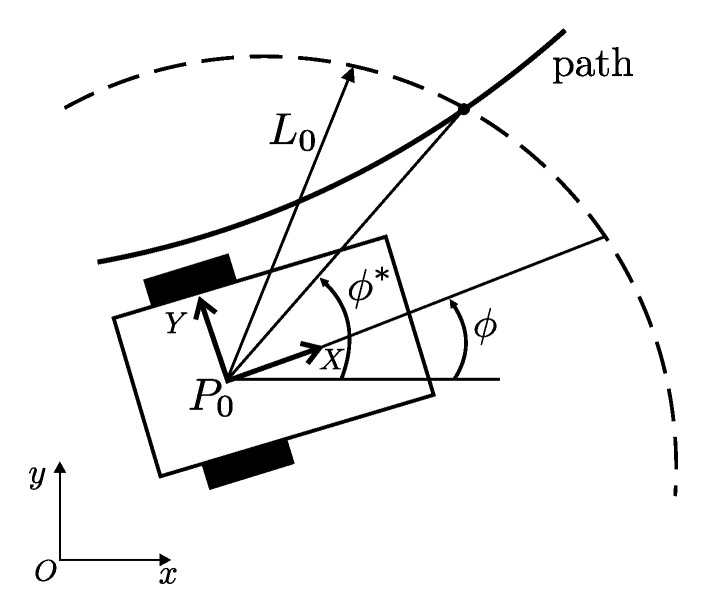
Path tracking task.

**Figure 4 sensors-20-06522-f004:**
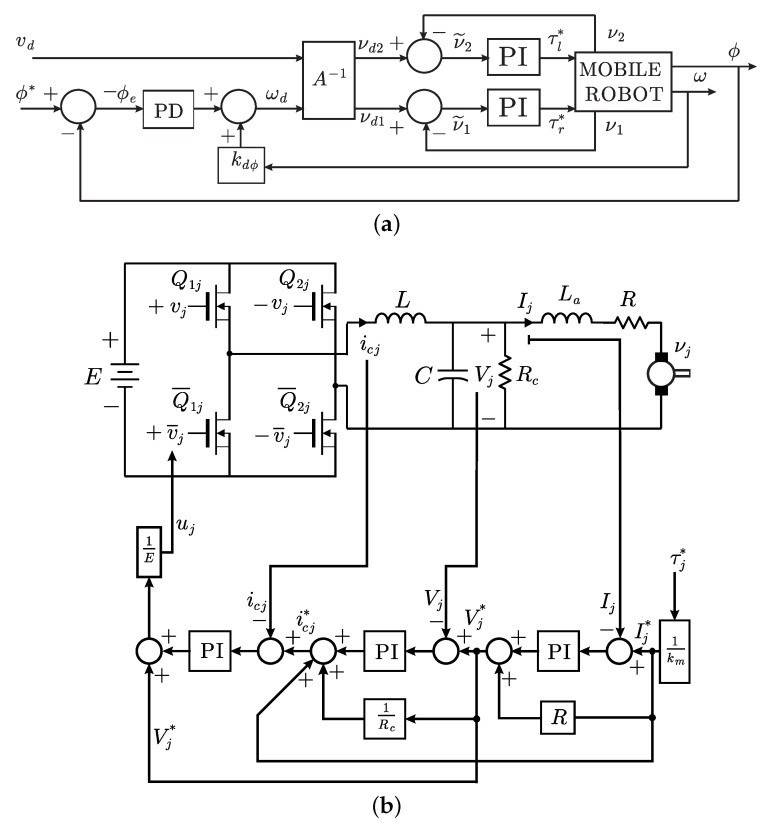
Block diagram of controller in Proposition 1. (**a**) PD path-tracking external loop and PI velocity loop. (**b**) Block MOBILE ROBOT in Figure 4a. Symbol $\tau_{j}^{*}$ stands for either $\tau_{r}^{*}$ or $\tau_{l}^{*}$.

**Figure 5 sensors-20-06522-f005:**
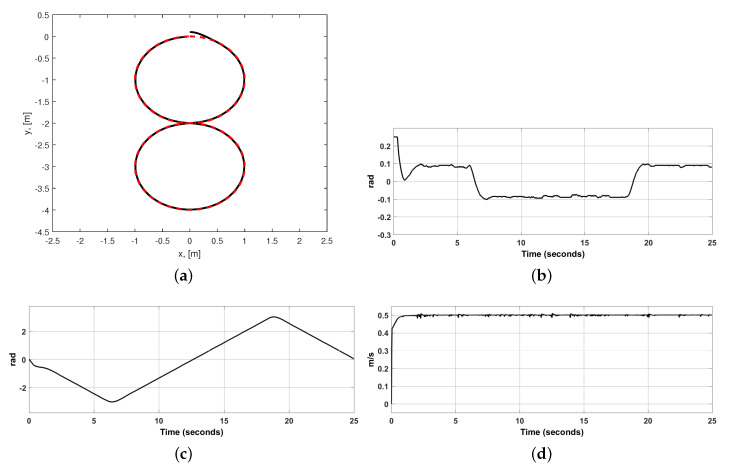
Path-tracking variables. (**a**) Dashed: desired path. Continuous: actual WMR position. (**b**) Tracking error $\phi_{e}$. (**c**) Actual WMR orientation ϕ. (**d**) Translational velocity *v*.

**Figure 6 sensors-20-06522-f006:**
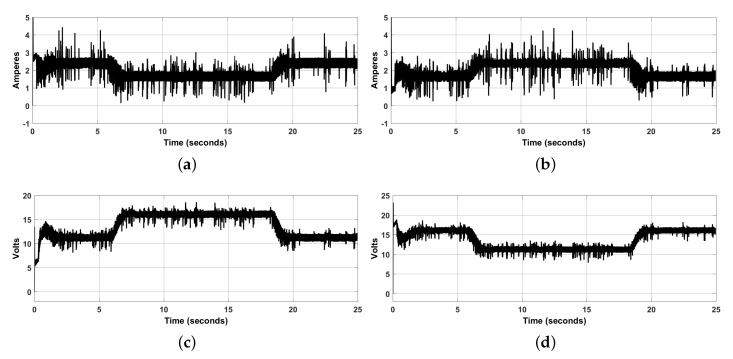
Power electronic converter variables. (**a**) Electric current in the Buck converter inductor no. 1 (right), $i_{c1}$. (**b**) Electric current in the Buck converter inductor no. 2 (left), $i_{c2}$. (**c**) Voltage at the Buck converter capacitor no. 1 (right), $V_{1}$. (**d**) Voltage at the Buck converter capacitor no. 2 (left), $V_{2}$.

**Figure 7 sensors-20-06522-f007:**
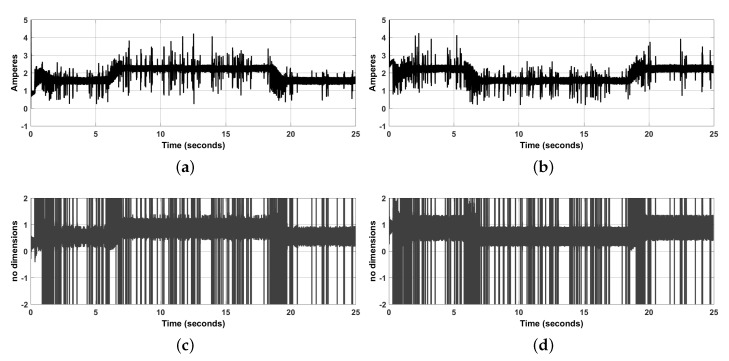
Electric current through the motor armatures and transistors on-off average signals. (**a**) Electric current in the right DC-motor, $I_{1}$. (**b**) Electric current in the left DC-motor, $I_{2}$. (**c**) On-off signal in the right Buck converter transistor, $u_{1}$. (**d**) On-off signal in the left Buck converter transistor, $u_{2}$.

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
