# Peer review of "Path-Tracking of a WMR Fed by Inverter-DC/DC Buck Power Electronic Converter Systems"

_sensors, 2020, doi:10.3390/s20226522_

Round 1

Reviewer 1 Report

Nice piece of work, well edited and especially the chapter with the mathematics equations is very sufficient. In terms of reference support the article is considered weak and this needs to be improved. There are less than 10 references out of 27 that comes from a period of the last 5 years. In such highly technical research, it would be very effective for your work to update the reference list with more up to date relevant research. In addition, try to modify the introductory part and keep a serial way of including the references in the main text, it is not appropriate to include mixed references in the text and not in an numbered serial order .

Reviewer 2 Report

This paper presents a trajectory tracking control algorithm for WMR fed by inverter-DC/DC Buck power converter system. For me, this paper seems to focus on the modeling of the motor dynamics and their power electronics systems, and just take the trajectory tracking of WMRs as a case study. Following are my comments:

1- The Literature review is very weak. Authors have to improve it considering the related works in the literature for the field of trajectory tracking of WMRs.

2- Trajectory tracking control was extensively studied in the literature. What are the main contributions compared to the related works in the literature. Authors mentioned their main contributions, but they do not compare it to the existing works.

3- PID control to solve the trajectroy tracking of WMRs in already exist in the literature. See the following refs for example:

a) Yingchong Ma, Gang Zheng, Wilfrid Perruquetti, Zhaopeng Qiu, "Motion planning for non-holonomic mobile robots using the i-PID controller and potential field", Intelligent Robots and Systems (IROS 2014) 2014 IEEE/RSJ International Conference on, pp. 3618-3623, 2014.

b) Y. Ma, G. Zheng, W. Perruquetti and Z. Qiu, "Motion planning for non-holonomic mobile robots using the i-PID controller and potential field," 2014 IEEE/RSJ International Conference on Intelligent Robots and Systems, Chicago, IL, 2014, pp. 3618-3623, doi: 10.1109/IROS.2014.6943069.   So, what are the contributions of this works compared to the related works?

4- Based on the problem statement mentioned in Page 5, the trajectory tracking problem is just the error difference in the orientation angle between the reference and actual trajectory. This is completely wrong, as the tracking error has to include the error in position (x,y) and orientation. Authors have to read many papers related to the trajectory tracking control of WMRs.

5- As this work is based on simulations, it is important to compare the proposed control algorithm with another algorithms in the literature. 

Reviewer 3 Report

The paper treats the path-tracking control of a wheeled mobile robot driven by two permanent magnet brushed DC motors which are fed by two DC/DC inverter.

I have the following remarks:

  • The authors develop a complex mathematical model of the analyzed system.

·         Rigorously justified theoretical control solutions are proposed.·         I notice an obvious inequality between the theoretical part, very widely presented, and the experimental one, much more modest.·         The experimental results must be presented in more detail. The figures do not satisfactorily illustrate the suggested performance.·         I propose to accept the paper but I suggest the development of the experimental part.

Round 2

Reviewer 2 Report

After the modified version, all my comments are covered, except the last one. Authors have to compare their results with other results in the literature. Even if their new algorithm is new, many related work solving the path tracking problem of WMRs considering WMRs kinematics and dynamics as well as motors dynamics. See the following ref for example:

Martins, F. N.; Celeste, W. C.; Carelli, R.; Sarcinelli-Filho, M. & Bastos-Filho, T. F.
An adaptive dynamic controller for autonomous mobile robot trajectory tracking
Control Engineering Practice, Elsevier BV, 2008, 16, 1354-1363.

As you tested your algorithm in simularions, you have to compare it with another one, at lest to show that your algorithm is better and solve a real problem.
